# Mutational Hotspot in the SARS-CoV-2 Spike Protein N-Terminal Domain Conferring Immune Escape Potential

**DOI:** 10.3390/v13112114

**Published:** 2021-10-20

**Authors:** Slawomir Kubik, Nils Arrigo, Jaume Bonet, Zhenyu Xu

**Affiliations:** 1Data Science Department, SOPHiA GENETICS, La Pièce 12, 1180 Rolle, Switzerland; skubik@sophiagenetics.com; 2Data Science Department, SOPHiA GENETICS, Rue du Centre 172, 1025 Saint-Sulpice, Switzerland; narrigo@sophiagenetics.com (N.A.); jbonet@sophiagenetics.com (J.B.)

**Keywords:** SARS-CoV-2 genome, coronavirus, spike NTD, W152, viral evolution, neutralizing antibody, immune escape

## Abstract

Global efforts are being made to monitor the evolution of SARS-CoV-2, aiming for early identification of genotypes providing increased infectivity or virulence. However, viral lineage-focused tracking might fail in early detection of advantageous mutations emerging independently across phylogenies. Here, the emergence patterns of Spike mutations were investigated in sequences deposited in local and global databases to identify mutational hotspots across phylogenies and we evaluated their impact on SARS-CoV-2 evolution. We found a striking increase in the frequency of recruitment of diverse substitutions at a critical residue (W152), positioned in the N-terminal domain (NTD) of the Spike protein, observed repeatedly across independent phylogenetic and geographical contexts. These mutations might have an impact on the evasion of neutralizing antibodies. Finally, we found that NTD is a region exhibiting particularly high frequency of mutation recruitments, suggesting an evolutionary path in which the virus maintains optimal efficiency of ACE2 binding combined with the flexibility facilitating the immune escape. We conclude that adaptive mutations, frequently present outside of the receptor-binding domain, can emerge in virtually any SARS-CoV-2 lineage and at any geographical location. Therefore, surveillance should not be restricted to monitoring defined lineages alone.

## 1. Introduction

RNA viruses display particularly high mutation rates [1], with SARS-CoV-2 undergoing approximately 10^−3^ substitutions/site/year [2]. Globally, the selective pressure imposes conservation of adaptive mutations facilitating the viral spread. The overall success of viral transmission depends on the mutation rate, the extent of immune response, and the population size [3]. During the pandemic, where population size is large, rapid increase in the frequency of alterations is observed at critical positions of the viral genome. Two commonly reported forces shaping the natural selection for SARS-CoV-2 are the adaptation to host [4] and the evasion of the immune response [5], including immunity triggered by the vaccines [6]. Consequently, the evolutionary rate is particularly high for the S gene encoding the Spike protein [7], the main contact point with the ACE2 receptor of the host cell [8]. Importantly, Spike also serves as the immunizing agent in the majority of COVID-19 vaccines [9].

It is expected that mutations improving viral fitness emerge independently across unrelated viral clades. An example of an adaptive mutation that emerged relatively early during the pandemic is the D614G substitution in Spike, by the end of 2020 it was present in almost every SARS-CoV-2 genome in the world [10], and believed to improve the Spike trimer interaction with ACE2 [4,11]. Since the final months of 2020, increase in frequency of other mutations was observed, with N501Y and E484K being two prominent examples. The mechanisms by which they confer evolutionary advantage to SARS-CoV-2 vary. In particular, N501Y increases the adaptation to the host by enhanced binding to the ACE2 receptor [12,13,14], resulting in more efficient transmission [15]. In contrast, E484K appears as selectively advantageous by decreasing the strength of the interaction with neutralizing antibodies [5,16,17], which facilitates evasion of the immune response. More recently, L452R substitution was reported to have similar properties to E484K [5,16,18,19]. Importantly, these mutations have arisen repeatedly and independently within diverse, unrelated genomic contexts, and at distant geographical locations, being examples of convergent evolution. Moreover, it may be expected that certain genomic positions under strong negative frequency-dependent selections—as expected in the context of immunity-escaping processes [20]—will display a diverse spectrum of mutations.

Adaptive traits require close monitoring, particularly because they are likely to appear as increasingly prominent within SARS-CoV-2 strains under global vaccination efforts aimed at establishing herd immunity. Several studies focused on evaluating potential impact of mutations on the viral spread and antibody evasion [16,21,22,23,24,25,26,27]. Most investigations focused on the receptor binding domain (RBD) of the Spike, the immunodominant part of the protein [28] containing the ACE2-interacting interface. However, mutations at sites outside of the RBD, such as D614, might also have strong impact on both, the infectivity and immune escape. For example, the N-terminal domain (NTD) of the Spike was shown to be a potent target for neutralizing antibodies [6,29,30], particularly in a region referred to as the antigenic “supersite” [31,32].

By screening SARS-CoV-2 genome sequences for residues undergoing frequent and diverse mutations, we pinpointed W152, a residue present in NTD, whose alterations have the potential of being advantageous for viral transmission. We identified that several substitutions, leading to a limited set of amino-acid changes at position W152, were independently recruited numerous times across many distantly related phylogenetic contexts and diverse geographical locations, suggesting their adaptive character. Insights from structural studies confirm that the identified W152 substitutions remove an important interaction point for multiple potent neutralizing antibodies. Furthermore, we demonstrate that mutations in NTD were recruited more frequently than in other regions of Spike during the second wave of the pandemic, potentially due to improving viral fitness through the immune escape. Our work highlights the importance of monitoring individual mutations occurring outside of the Spike RBD.

## 2. Methods

### 2.1. Identification of Emerging Mutations in DDM Database

Aggregated statistics were collected from SOPHiA DDM database (https://www.sophiagenetics.com/technology/, as of 23 March 2021) (Appendix A). Briefly, genotypes were collected for samples with at least 98% SARS-CoV-2 genome covered, determined by taking into account mutations found at a fraction of at least 70% [33]. The country of sample origin was maintained for each genotype. Clades were assigned to the genotypes using Nextclade (version 0.13.0) [34] and lineages using Pangolin (version 2.3.2) [35]. The following aggregate statistics were collected for every detected mutation (including SNVs, insertions, and deletions): frequency in DDM, global frequency (based on data deposited in GISAID, https://www.gisaid.org), clades, and lineages in which the mutation was found, number of countries where a mutation was reported. We then filtered the dataset to keep genomic positions where (i) more than one type of mutation was reported; (ii) a mutation was present at least 5× more frequently in DDM than globally; and (iii) it emerged across multiple clades or lineages.

### 2.2. Analysis of Publicly Available SARS-CoV-2 Sequences

#### 2.2.1. Phylogenetic and Protein Data

We extended our analysis to publicly available data included in the Audacity global COVID phylogeny along with all Spike protein sequences (1,028,876 entries-spikeprot0406.fasta) deposited in GISAID as of 11 April 2021. Protein sequences were aligned against the Spike reference (YP_009724390.1 as obtained from NCBI, as of 11 April 2021), using muscle v3.8.31 [36] with default parameters for protein analysis and converted into VCF files using custom R scripts [37]. The Audacity phylogeny and VCF files were then merged to obtain a phylogeny of 566,422 tips (391,504 internal nodes) with Spike protein information available for all tips.

#### 2.2.2. Identification of Independent W152 Recruitments

Our analysis aimed to inventory independent recruitments of Spike mutations. From a phylogenetic standpoint, the task required regrouping SARS-CoV-2 genomes holding a Spike mutation of interest into sets of genomes that shared a common ancestor (i.e., “clades”). Assuming rare recombination, the most recent common ancestor (i.e., “mrca”) of a clade marked the “recruitment event” at which the mutation of interest arose in the tree. The remainder of the clade then replayed transmission of the mutant to new hosts and the creation of a contagion cluster.

Because the size of the Audacity tree rendered the most ancestral character reconstructions intractable, we opted for an ad-hoc heuristic that iteratively delineated clades and identified the respective mrca of mutation carrying sequences. To this end, we applied a tree walk algorithm that identified clades given a tree topology and a set of tips states. Our heuristic proceeded as follows (see Appendix A): (i) identify all tree tips containing a mutation of interest (i.e., “mutant tips”); (ii) for each mutant tip, move up to the parent node and collect all of its descending tips; (iii) calculate the proportion of retrieved tips that bear the mutation of interest (tips with missing data and tips already visited by the heuristic were excluded from the computation); (iv) if the proportion exceeded 0.95, then accept the parent node so as to increment the focal clade with a set of genomes that consistently contain the mutation of interest—while still allowing for punctual reversions to WT genotypes—and repeat steps (ii) to (iv). Keep ascending the tree until reaching a parent node where the threshold is not met. Once stopped, define the last visited node as the mrca and gather all descendant tips. The tree walk was initiated from every mutant tip and returned a collection of clades that was finally refined by merging clades that shared at least one genome.

The independent clades of W152 mutations were then characterized according to: amino acid change, clade size (number of genome occurrences belonging to the clade), earliest observation (deposition week, taken as a proxy of recruitment time), geographical spread (number of countries in which a clade was found), and status of concerning RBD mutations (i.e., L452R, E484K, N501Y) at the recruitment event (inferred by checking for the presence of RBD mutations shared within genomes closely related to the clade of interest). Finally, the analysis was extended to all Spike positions to establish the overall distribution of recruitment events of Spike mutations, as a function of time. The R packages ape v5.0 and fishplot v.0.5.1 were used to perform all phylogenetic analyses. All scripts were deposited on GitHub.

### 2.3. Structure-Based Prediction of the Impact of the Mutations on Spike Interactions

We assessed the effect of W152 mutants on neutralizing antibody (nAb) recognition by generating single point mutants (W152C, W152L, and W152R) and evaluating their changes in binding free energy (ddG) against 5 different NTD-target antibodies (1-87 and 5-24 [32], 4A8 [29], FC05 [38] and S2X333 [6], structures obtained from the Protein Data Bank [39]) using Rosetta [40]. nAbs were selected based on the availability and interaction angle [32] to provide a broad view of the possible scenarios. For each experiment (mutant-antibody pair), side chain minimization was performed after the mutation and before the ddG analysis. As minimization in Rosetta is a stochastic-based process, a total of 100 decoys were generated for each experiment to define a distribution of ddG values. Finally, all decoys of ddG (regardless of the mutation) for a given antibody were normalized to the distribution obtained with wild type (WT) Spike for that antibody.

## 3. Results

### 3.1. Tryptophan at Position 152 Is a Mutational Hotspot

We screened SARS-CoV-2 genomes present in the DDM database (see Methods section) in order to identify novel, potentially concerning mutations within the S gene, defined as (i) multiple non-synonymous substitutions present at a single position; (ii) displaying increased frequency in comparison with global frequency; (iii) independent recruitments across multiple lineages; and (iv) across multiple geographical locations. This approach identified distinct mutations at position W152 of the Spike NTD resulting in substitution of tryptophan to leucine (W152L) or arginine (W152R) (Table 1). Moreover, two other (less frequent) substitutions at the same position were identified in DDM: cysteine (W152C) or glycine (W152G). Importantly, both W152L and W152R were reported across unrelated phylogenetic contexts (i.e., clades and lineages) and at distant geographical locations. Of note, W152C is a clade-defining mutation for the CAL.20C clade (B.1.429 lineage) responsible for an outbreak in Southern California [41]. Frequent and diverse mutations at a single residue, emerging repeatedly and independently, suggested their putative adaptive role.

### 3.2. Diverse W152 Mutations Were Recruited Independently in Multiple Phylogenetic Contexts

We investigated the recruitment dynamics of W152 mutations in global datasets deposited in GISAID database. Due to a relatively low number of depositions in weeks 76–80 of the pandemic, compared to the preceding weeks (Appendix A), we only considered depositions with collection dates up to week 75. Rapid and steady growth was observed in the number of submitted sequences bearing three W152 substitutions (W152C, W152L, and W152R) during the second wave of the pandemic, in the period between December 2020 and April 2021 (weeks 56–75 of the pandemic) (Figure 1A,B). These substitutions were associated with 171 independent recruitments since the beginning of the pandemic (Figure 1C), with only sporadic cases (14.6%, 25/171) reported during the first wave (until week 55—Figure 1D). The largest cluster was reported for W152C with over 13,000 occurrences in 30 countries (including the CAL.20C lineage, referred to as the “California variant”), with the second-largest containing almost 1500 sequences bearing W152L and present in 20 countries. However, most clusters were relatively small in size (≤5 sequences reported for 86% clades [147/171]) and present in only one country (Appendix A). This observation pointed to frequent and independent recruitments rather than spreading of viruses bearing W152 mutations due to cross-border transmission. During weeks 65–75, the number of independent W152 mutation recruitments was in the range between the 90th and 99th percentile among independent mutation recruitments reported for all Spike positions (Figure 1E). These observations placed W152 as one of the most dynamic Spike positions in terms of mutation recruitments, just behind N501, E484, and ahead of L452 (Appendix A). W152 was also one of only two Spike residues with three independent substitutions having at least 500 occurrences in GISAID being reported for each, with another NTD residue (D80) being the other one (Appendix A).

In order to investigate whether W152 substitutions might confer direct evolutionary advantage to SARS-CoV-2, we investigated Spike mutations most frequently co-occurring with each W152 substitution in individual sequences (Figure 2A). The only Spike mutation co-occurring with all three tested W152 substitutions was the ubiquitous D614G. Each substitution displayed relatively high frequency of different co-occurring Spike mutations. For W152C, the most frequent were S13I and L452R, both clade-defining for the 20C/S:452R clade (lineage B.1.429). In case of W152L, these were E484K and G769V (found in R.1 lineage) and for W152R–HV69del, Y144del, N501Y, A570D, P681H, T716I, S982A, and D1118H, defining for clade 20I/501Y.V1 (lineage B.1.1.7). When frequency of co-occurring mutations was calculated, giving equal weight to each clade, irrespective of the number of sequences within, we observed that, for most clades (67.8%, 116/171), W152 substitution did not combine with any of the prominent RBD mutations suspected of being advantageous (Figure 2B,C). However, sequences in each of the largest clusters for individual substitutions contained at least one of these: L452R (for W152C), E484K (W152L), or N501Y (W152R) (Figure 2C). After excluding the single largest cluster for each substitution, the fraction of sequences without adaptive RBD mutation was 64%. Importantly, our phylogenetic analysis indicated that the ancestor clades of each of the largest clusters were already bearing RBD mutations when acquiring W152 substitutions (see Section 2.2.2). The above observations point to a potential key role of the tryptophan at position 152 for SARS-CoV-2 fitness, regardless of the co-existing adaptive RBD alterations.

### 3.3. W152 Is an Important Interaction Point for Neutralizing Antibodies

Distinct regions of the Spike are subject to different evolutionary constraints. RBD domain mutations can be steered by both: increased ACE2 binding and neutralizing antibody (nAb) evasion while the NTD alterations are mostly determined by the nAb escape potential. We sought to investigate how W152 mutations affect the recognition of the NTD domain by selected NTD-targeting nAbs. To this end, we introduced each of the mutations of interest (W152L, W152R, and W152C) into 5 Spike NTD structures, each bound to a different NTD-nAb (see Methods section), performed side-chain optimization, and ddG analysis (Figure 3A). In all cases, we observed a residue in the Ab chain engaged in a pi stacking interaction with W152 in the wild type Spike protein (Figure 3B). Moreover, 1–87 and 4A8 nAbs wrap W152 inside one of the CDR loops, making it a key position contributing to the interaction. In 5–24, the position is located just outside of the interface, but close enough for the minimal conformational changes to allow it to participate in the binding. FC05 and S2X333 use W152 for secondary interactions; thus, the contribution of the residue to the interface is negligible and its mutation can be easily compensated by a side chain movement.

The stacking interactions were lost with any of the considered mutations, thus effectively decreasing the affinity of the nAb to bind to NTD (Figure 3A). As expected, the effect varied depending on the extent of the W152 participation in the main interacting surface. Thus, in the case of the nAbs 1-87 (Figure 3C) and 4A8 (Figure 3D), a significant drop in affinity was observed, while in peripherally interacting nAb, such as 5-24, FC05, and S2X333 the effect was little or negligible (Figure 3A). In case of 1-87 and 4A8, the amino acid substitution not only results in the loss of the pi stack interaction, but also affects the pocket generated by the antibody’s CDR loop, which leads to a substantial loss of the binding affinity. The effects are especially drastic for W152L due to the exposure of the hydrophobic side chain.

NTD mutations are enriched in the second wave of the pandemic and point to immune escape potential. The W152 residue is placed in the vicinity of mutations of the NTD domain that received increasing attention owing to recent emergences in the variants of concern (Figure 4A, L18, H69, or Y144). NTD constitutes an exposed part of the Spike protomer, making it a prominent target for antibodies, yet, contrary to RBD, mutations within NTD have potentially little impact on receptor binding. We investigated the possibility that residues present in NTD undergo extensive mutagenesis, facilitating immune escape without hampering the interaction with ACE2. During the first wave of the pandemic (until week 55), mutation recruitments were distributed relatively uniformly across Spike domains (Figure 4B). On the contrary, during the second wave (i.e., week 56 and onwards), NTD displayed an elevated number of mutation recruitments compared to other parts of the protein (Figure 4C). This localized bias in diversity was statistically significant (Figure 4D) and could result from adaptive changes in response to global immunity. The number of mutation recruitments per position correlated significantly with the evolutionary lability of the Spike protein observed across *Coronaviridae* (Appendix A; *p* < 10^−5^ for each tested region). Nevertheless, recruitment events were significantly more common for the NTD in comparison to RBD and the remainder of the Spike protein residues, confirming similar evolutionary pattern for the related viruses. These observations suggest that globally, NTD mutations have higher propensity of being advantageous than alterations in other regions of the protein.

## 4. Discussion

Accumulating body of evidence suggests a key role of SARS-CoV-2 Spike NTD mutations in viral evolution of the virus. Based on conservation estimates and the number of independent mutation emergences, we demonstrate that this domain undergoes more rapid evolutionary changes in comparison with other parts of the protein. This localized evolution gained momentum during the second wave of the pandemic. It is possible that this trend emerged in response to global increase in immunity. The most prominent forces driving the selection of Spike mutations are the increased interactions with ACE2 and the evasion of neutralizing antibodies (nAb). NTD and RBD constitute the most exposed parts of the Spike making them the most likely targets of the immune response. As demonstrated by us and others, mutations in these domains often facilitate immune evasion [5,6,16,17,21,22,25,32]. However, RBD mutations are more evolutionarily constrained due to their impact on interaction with ACE2. Given the relatively small contribution of NTD to ACE2 binding, alterations in this domain might constitute the evolutionary ‘disguise’ the virus uses to avoid antibody neutralization. The variability is not restricted to amino acid substitutions as a significant number of deletions was also reported in NTD and linked to the immune escape [42]. Identification of potential nAb that can be used against different variants of RBD and NTD is of great importance, considering that antibody cocktails might act collaboratively to impede the progression of viral infection [38,43].

Another possibility explaining increased rate of NTD mutation emergences is that this domain facilitates the entry into the host cell through interactions with membrane glycan residues or one or more of putative co-receptors [44]. It is supported by the observation that many mAbs targeting the NTD can specifically inhibit the ACE2-independent cell entry [45]. In this perspective, the emerging NTD mutations could have positive impact also on the efficient host invasion, similarly to adaptive RBD mutations.

We identified W152 as an NTD residue undergoing particularly extensive evolutionary dynamics, highlighted by multiple individual substitutions emerging across many phylogenetic and geographical contexts. Remarkably frequent mutation recruitment events were reported at this position globally, with a clear increase in intensity since the end of 2020 (week 55 of the pandemic). However, the genetic diversity reached by the virus in the second half of 2021 imposes considering the adaptive potential of mutation combinations rather than that of individual alterations. Even if the vast majority of recruitment events for W152L, W152R, or W152C was not accompanied by the presence of another particular mutation, the largest clusters were characterized by the co-recruitment of one of the well-described adaptive RBD mutations (E484K, N501Y, or L452R, respectively). Although the contribution of W152 mutations to those three events could not be simply decoupled from that of their co-occurring alterations in RBD, the adaptive potential might have been increased by their co-occurrence. In line with this finding, a recent study demonstrated that W152C allows to further increase B.1.429 lineage infectivity in comparison to L452R alone [18]. Curiously, the adaptive potential of mutations is not always additive. A recent study shows that resistance conferred by E484K substitution was reversed by the addition of other alterations [46]. This observation might explain why mutations showing adaptive potential in isolation might not achieve expanded spread when recruited in certain genomic context. It also points to the increasing complexity of emerging viral genotypes as a growing challenge for predictions of the transmissibility.

It is generally appreciated that advantageous mutations initially arise as the quasi-species, present only in a fraction of viral genomes within a given host. Providing a competitive edge—they are progressively increasing in prevalence and are eventually transmitted to new hosts, giving rise to new clades responsible for infection clusters. In this regard, the advantage conferred by W152 mutations might be exemplified by a reported increase in the intra-host fraction of genomes bearing W152L substitution during the infection [47].

Tryptophan at position 152 is located in the antigenic “supersite” frequently targeted by nAbs [31,32]. Our analysis suggests that mutations of W152 diminish the strength of interaction with several antibodies, consistent with other reports [6,29,32,48]. As the chemical properties of each side chain of the amino acids present in the mutants differ (leucine–hydrophobic, arginine–basic, cysteine–sulfide) the advantage might stem from the fact of removing tryptophan (aromatic) from the relevant position rather than establishing specific novel interactions. Such alteration decreases the propensity of forming stable stacking interactions, similar to those observed in the 1–87 or 4A8 antibody complexes with Spike [29]. The “supersite” can be easily distorted by mutations [6,49]. It was demonstrated that W152 in the Spike of the Delta variant is significantly shifted comparing to its position in the wild type protein, which probably contributes to the complete loss of reactivity to anti-NTD neutralizing antibodies in that strain [50]. Anti-NTD antibodies contacting W152 were also shown to be inefficient in the W152C-bearing Epsilon variant [48] and against Spike with W152R substitution [43]. Alarming evidence supporting the role of W152 in evading immune response comes from a report of the R.1 lineage, bearing W152L [51], being responsible for a local outbreak in a population having underwent a vaccination program [52].

Although any SARS-CoV-2 evolution study during the pandemic is sensitive to the time of analysis, the conclusions emerging from our work suggest that large emphasis should be put on monitoring alterations occurring in NTD. Significant efforts are spent on tracking the spread of specific variants of concern, such as the B.1.1.7 (Alpha) or the B.1.167.2 (Delta). However, our study outlines the importance of monitoring the emergence, recruitment, and spread of individual mutations and their combinations outside of the boundaries set by the lineage assignment. Residue mutations, such as L18, H69, Y144, W152, L452, E484, or N501 occurred frequently, and have been recruited across many independent SARS-CoV-2 lineages. As of writing, the W152 mutations spurred 171 independent clusters, inferred from the number of mutation recruitments, which contributed to infection of at least 15,000 patients in total. Hence, monitoring efforts should not overlook the fact that evolutionary advantageous mutations can emerge in virtually any SARS-CoV-2 lineage and at any geographical location.

## Figures and Tables

**Figure 1 viruses-13-02114-f001:**
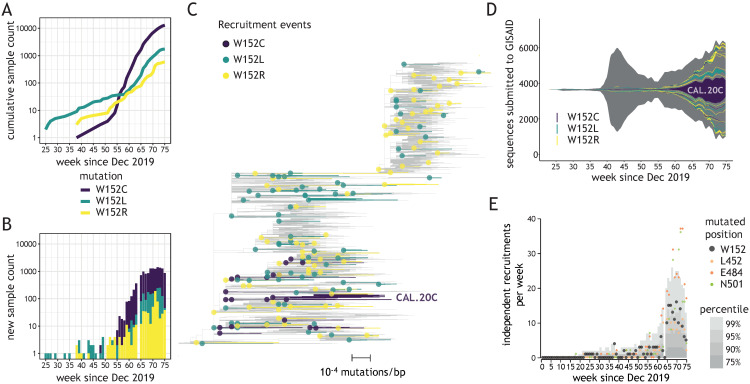
Diverse W152 mutations emerged independently across numerous genomic contexts. (**A**) A cumulative number of sequences bearing three W152 mutations uploaded to GISAID (y-axis) depending on the upload date (x-axis). (**B**) Number of sequences bearing three W152 mutations uploaded weekly to GISAID (y-axis) depending on the upload date (x-axis). (**C**) Representative phylogenetic tree of analyzed sequences (displaying 1% of the global Audacity tree); 171 independent W152C/L/R clades are identified in the present study and displayed to outline mutation recruitment events (dots) and ensuing contagion clusters (edges); largest cluster for W152C (CAL.20C) is indicated. (**D**) Fish plot of all SARS-CoV-2 genomes deposited in GISAID, with color-coded areas corresponding to the numeric abundance of clades that recruited W152C (purple—20 events), W152L (turquoise—75 events) or W152R (yellow—76 events) mutations, as a function of the upload date. (**E**) Plot showing the weekly count of recruitments of clades bearing mutations at positions W152 (black), L452 (orange), E484 (red), and N501 (green); grey areas indicate values for percentiles 75th, 90th, 95th, and 99th of mutation recruitments across all Spike positions.

**Figure 2 viruses-13-02114-f002:**
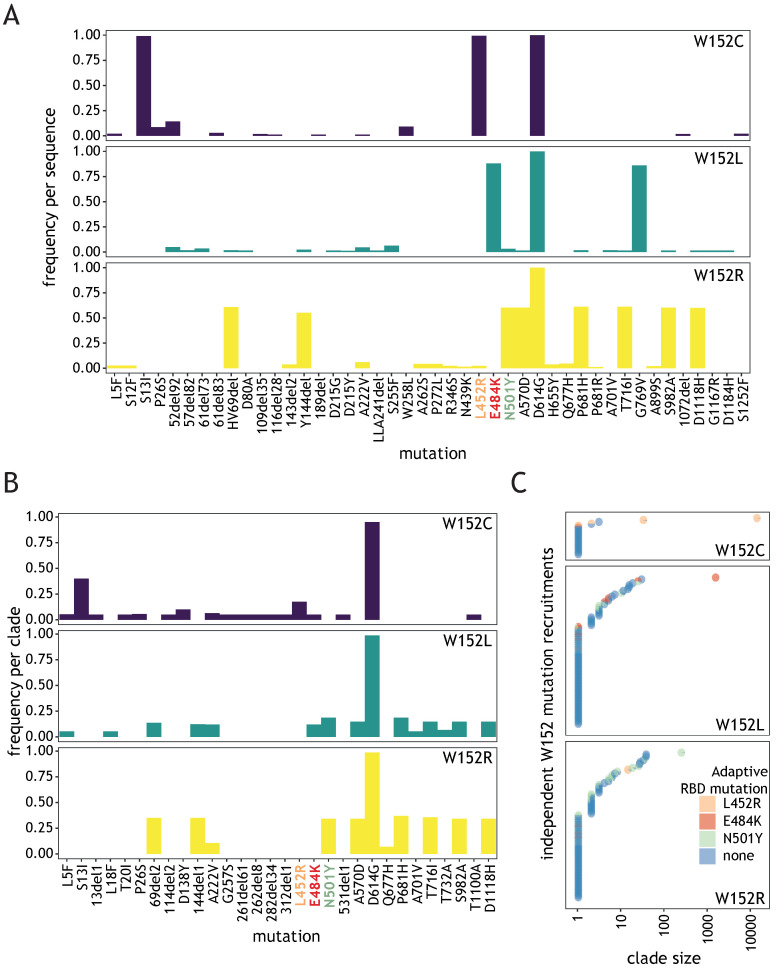
Most W152 mutation recruitments are not associated with a known adaptive RBD mutation. (**A**) Plots showing, for the three W152 substitutions, the frequency of co-occurrence of non-synonymous Spike mutations per individual sequence, present at a frequency of at least 0.01; known, adaptive RBD mutations are indicated by orange, red, and green labels. (**B**) As (**A**), but calculated per clade, regardless of clade size, shown for frequencies of at least 0.05. (**C**) Plot showing, for three W152 substitutions, independently recruited clades (circles) colored depending on the co-existing adaptive RBD mutation and positioned depending on the clade size (x axis).

**Figure 3 viruses-13-02114-f003:**
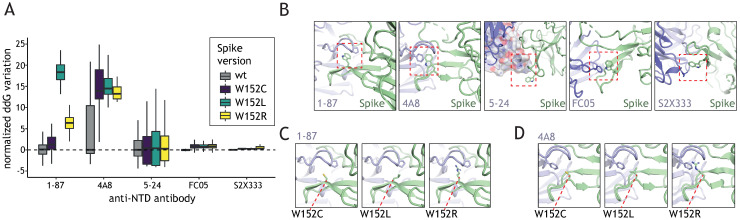
Interactions with neutralizing antibodies are weakened by W152 mutations. (**A**) Normalized ddG calculation with the wild type Spike and the three W152 mutants (W152C, W152L, W152R) for binding to different anti-NTD nAb, based on available structures. For complexes with 1–87 and 4A8, W152 is found within the binding interface, for 5–24, FC05 and S2X333 complexes W152 is a secondary interaction point at the edge of the binding interface. (**B**) a zoomed-in view of the W152-containing region for available structures of Spike in complex with anti-NTD nAbs; in all cases, the nAb presents a residue engaged in pi stacking with W152 (red box); aromatic amino acids are the binding partners for W152 in all nAbs except S2X333, where arginine is involved in the interaction. (**C**,**D**) Interaction interfaces of Spike W152 mutants with 1–87 and 4A8, respectively.

**Figure 4 viruses-13-02114-f004:**
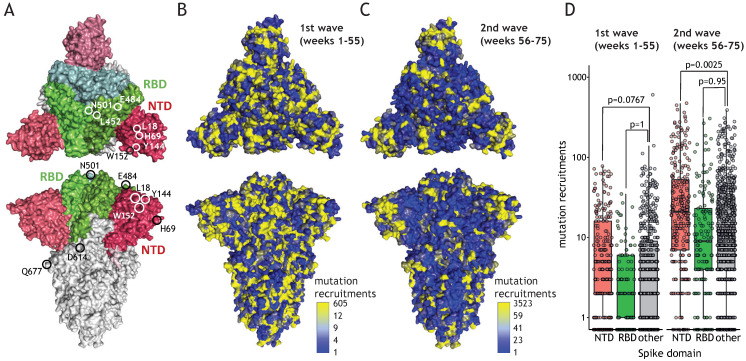
Extensive NTD evolution during the second wave of the pandemic. (**A**) Structural model of Spike trimer (PDB Id: 7DDD); RBD marked in green, NTD marked in red; chosen mutation positions are indicated. (**B**) Spike structure model colored according to the number of mutation recruitments for each amino acid position for weeks 1–55 of the pandemic. The color ramp partitions the number of recruitments into eight equally-spaced bins ranging between the minimum value (blue) and two times the average (light blue); all remaining observations that include more variable amino acid positions are color-coded in yellow. (**C**) As B, but for weeks 56–75. (**D**) Boxplot showing mutation recruitment events for all Spike positions present in NTD (red), RBD (green), or other parts of the protein (grey) for weeks 1–55 (left) or 56–75 (right); the statistical significance is assessed using permutation testing (10,000 permutations).

**Table 1 viruses-13-02114-t001:** Aggregated data on W152 mutations identified in DDM database.

Mutation	DDM Frequency (%)	Global Frequency (%)	Country Count	Continent Count	Clade Count	Clades	Lineage Count	Lineages
Trp152Leu	0.82	0.06	3	2	5	19A, 20A, 20B, 20I/501Y.V1, 20H/501Y.V2	9	B, B.1, B.1.1, B.1.1.7, B.1.29, B.1.351, B.1.416, B.1.420, R.1
Trp152Arg	0.73	0.04	3	2	4	19A, 20A, 20D, 20I/501Y.V1	7	B, B.1, B.1.1.1, B.1.111, B.1.1.7, B.1.166, B.1.402
Trp152Cys	0.02	1.56	1	1	1	20C	1	B.1.429
Trp152Gly	0.02	2 × 10^−4^	1	1	1	20I/501Y.v1	1	B.1.1.7

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
