# Peer review of "Mutational Hotspot in the SARS-CoV-2 Spike Protein N-Terminal Domain Conferring Immune Escape Potential"

_viruses, 2021, doi:10.3390/v13112114_

Round 1
Reviewer 1 Report
Kubik et al. investigated the patterns of Spike mutations to identify mutational hotspots and their impact on SARS-CoV-2 evolution. Overall, the manuscript is informative, which provides new insight on how the W152 substitutions in the N-terminal domain (NTD) of the Spike protein conferring immune escape potential. However, there are some concerns.
- How the aggregated statistics were collected from SOPHiA DDM database is not clear. Please explain it adequately so that everyone can access it.
- Please generate a schematic diagram to demonstrate each step performed in the methods and their significance.
- The literature survey is not up-to-date. Please add more references in the Introduction and Discussion sections.
- It will be interesting to check if the W152 substitutions can alter the conformational epitope.
- The discussion section should be re-written properly. Highlight how the W152 mutations can characterize the ‘variants of concern’, including the delta-variant.
Reviewer 2 Report
The manuscript by Kubik et al. aims to spot advantageous mutations emerging independently and to evaluate their impact on SARS-CoV-2 evolution and evasion of neutralizing antibodies. The authors focused on the hotspot of Spike NTD and specifically the W152 with respect to ACE2 binding and protein flexibility. The authors also suggest that it is yet another adapted mutation outside of the RBD.
Major comments:
- There are numerous reports that indicate W152 as an important site for antibody evasion some were published in Journals, others are in BioRxiv. Changes in the NTD were studied, with detailed neutralization assays were conducted. An example is for the antibody 4A8 and more (listed as Ref 29). Another detailed research (Ref 6) addresses the role of W152: “the W152C mutation reduced recognition of six NTD neutralizing mAbs, including a complete loss of binding for two of them”. It was further suggested (based on MS support) that the formation of disulfide cysteine bond (in B.1.427/B.1.429) between C136 and W152C. So in this manuscript, the importance of W152C in antibodies evasion was validated (based on energy calculation). With such previously reported analysis, novelty is modest.
- At present and during waves 3 and 4, amino acid changes in W152 were basically disappeared (globally) while other ‘adapted’ mutations (in RBD) took over the viral population. What is the evidence that it is indeed adapted and related to selective pressure of antibodies or immunization?
- The 3D model presented ignores neighboring glycans. It will be useful to address the changes in view of the surrounding glycan moieties that probably affect antibodies accessibility.
- Although the thresholds and filters were described with great details (and source code is shared), it is not fully clear whether W152 was the only one that passed the set of thresholds or was it selected as a showcase. In any case, listing other NTD amino acids changes that are evolved independently (e.g. amino acid 138, 157, 144 etc) is valuable. Moreover, prevalent adapted sites are ‘deleted’ (e.g. 156, 157). This issue of deleted amino acids should be addressed (in discussion). It will help to reply -in what sense W152 is more adapted than others.
- While any study on the Covid-19 pandemic is sensitive to the time of analysis, the pandemic took a new route (after April 2021), with the delta variant overrides others. In this view, L482R keeps dominating the current pandemic (after April 2021). It is clear that the presented analysis is time-sensitive. Whether the conclusions hold with current is an important discussion.
- The discussion of whether preexisting adapted mutations provide advantages is a bit unclear. If each of the 3 reported ‘adapted’ mutations plays the same fitness improvement as does W152 change, it would expect them to evolve together. Please explain.
- 3A shows the sensitivity of the energy calculation. Showing results for another co-occurring mutation is useful. Als the changes in M18 or T19, 144 could be used for comparison.
- Some sentences call for rephrasing (by evidence provided that support such statements)
- In Discussion: “undergoes more rapid evolutionary changes in comparison with other parts of the protein” cannot be generalized as stated. The long-term evolutionary changes (when different coronaviruses and different hosts are compared) highlight the RBD and many of the ORFs and less so for the NTD.
- “likely in response to a global increase in immunity” is one of a number of possibilities and clearly was not shown to be the case.
Minor comments:
- Please put context to the statement “171 independent clusters that directly or indirectly contributed to contaminate over 15,000 patients”. How many ‘clusters? how many patients?
- Referring to “supersite” in NTD (that includes W152) was not mentioned.
- NTD was postulated in recognition of membrane glycan and other putative receptors. Is it likely to be a driving force for cumulative mutations for Wave 2?
- Figure 2 is graphically poor (unclear format). Fig 2B seems not so informative and can be removed.
- The References numbering is to be corrected. No need to write 1. (1)
- Figure 4D. “other’ is defined only in supplemental Fig. Please add to Fig. legend
- The text does not refer to any of the Suppl. Figures. The analysis is well explained in the supplemental.
Reviewer 3 Report
The authors proposed the role of mutations in the N-terminal domain of spike glycoprotein which can confer resistance to neutralizing antibodies. Even though the authors greatly analyzed the occurrence of NTD 152 substitution across the world by bioinformatic analysis, functional assessment of the predicted mutations has to be evaluated in vitro.
Major comment
The immune escape nature of the NTD mutants was predicted using binding analysis, which has to be validated using functional studies either by using recombinant mutant proteins or pseudotyped viruses bearing the spike mutations. The bioinformatic predictions may not strongly correlate with the in vitro or in vivo neutralizing antibodies against spike variants.
There are several mutations found in the N terminal domain of Spike but it is not clear from the text; how do the authors end up only in W152 mutant? Why not others?
The W152 mutation is found in all clades of SARS-CoV 2, but not all these viruses are highly pathogenic or highly transmissible. If such a case, how can the authors claim that these W152 immune escape variants?
The N terminal domain may have a role in virus entry. Recently Kartika et al. showed that the Cellular Entry Is Independent of the ACE2 Cytoplasmic Domain Signaling. Cells. 2021 Jul 17;10(7):1814. doi: 10.3390/cells10071814. Similarly, Cerutti G, et al. showed Potent SARS-CoV-2 neutralizing antibodies directed against spike N-terminal domain target a single supersite. Cell Host Microbe. 2021 May 12;29(5):819-833.e7. doi: 10.1016/j.chom.2021.03.005. This evidence suggests that the N terminal domain of SARS-CoV-2 may bind to another co-receptor to enter the host cells. Detailed discussion may enhance the visibility of the paper.
Round 2
Reviewer 1 Report
The authors significantly improved the manuscript.
Reviewer 2 Report
The authors addressed most comments by providing the rationale, changed the discussion to include some reservations (due to the changes in trends over time), and validates some of the statements.
Figure 2 is still fuzzy (technically), if possible, please improve
I appreciate the effort the author took to better explain the novelty and to justify the strength of their approach.